# Radiotherapy-Induced Changes in the Systemic Immune and Inflammation Parameters of Head and Neck Cancer Patients

**DOI:** 10.3390/cancers11091324

**Published:** 2019-09-06

**Authors:** Katalin Balázs, Enikő Kis, Christophe Badie, Enikő Noémi Bogdándi, Serge Candéias, Lourdes Cruz Garcia, Iwona Dominczyk, Benjamin Frey, Udo Gaipl, Zsolt Jurányi, Zsuzsa S. Kocsis, Eric Andreas Rutten, Géza Sáfrány, Piotr Widlak, Katalin Lumniczky

**Affiliations:** 1Department of Radiation Medicine, National Public Health Center, 1221 Budapest, Hungary; balazs.katalin@osski.hu (K.B.); kise@osski.hu (E.K.); nbogdandi@hotmail.com (E.N.B.); safrany.geza@osski.hu (G.S.); 2Centre for Radiation, Chemical and Environmental Hazards, Public Health England, Chilton, Didcot, Oxfordshire OX11 0RQ, UK; christophe.Badie@phe.gov.uk (C.B.); lourdes.CruzGarcia@phe.gov.uk (L.C.G.); Eric.AndreasRutten@phe.gov.uk (E.A.R.); 3Université Grenoble Alpes, CEA, CNRS, IRIG, CBM, F-38000 Grenoble, France; serge.candeias@cea.fr; 4Maria Skłodowska-Curie Institute—Oncology Center, Gliwice Branch, 44-101 Gliwice, Poland; iwona.dominczyk@io.gliwice.pl (I.D.); piotr.widlak@io.gliwice.pl (P.W.); 5Department of Radiation Oncology, Universitätsklinikum Erlangen, 91054 Erlangen, Germany; benjamin.frey@uk-erlangen.de (B.F.); udo.gaipl@uk-erlangen.de (U.G.); 6Department of Radiobiology and Diagnostic Onco-Cytogenetics, Centre of Radiotherapy, National Institute of Oncology, 1122 Budapest, Hungary; juranyi.zsolt@oncol.hu (Z.J.); kocsis.zsuzsa@oncol.hu (Z.S.K.)

**Keywords:** radiotherapy, head and neck cancer, PBMC, immune phenotyping, regulatory T cell, dendritic cell, natural killer cell, plasma proteins, gene expression

## Abstract

Though radiotherapy is a local therapy, it has systemic effects mainly influencing immune and inflammation processes. This has important consequences in the long-term prognosis and therapy individualization. Our objective was to investigate immune and inflammation-related changes in the peripheral blood of head and neck cancer patients treated with radiotherapy. Peripheral blood cells, plasma and blood cell-derived RNA were isolated from 23 patients before and at two time points after radiotherapy and cellular immune parameters, plasma protein changes and gene expression alterations were studied. Increased regulatory T cells and increased CTLA4 and PD-1 expression on CD4 cells indicated an immune suppression induced by the malignant condition, which was accentuated by radiotherapy. Circulating dendritic cells were strongly elevated before treatment and were not affected by radiotherapy. Decreased endoglin levels in the plasma of patients before treatment were further decreased by radiotherapy. Expression of the FXDR, SESN1, GADD45, DDB2 and MDM2 radiation-response genes were altered in the peripheral blood cells of patients after radiotherapy. All changes were long-lasting, detectable one month after radiotherapy. In conclusion we demonstrated radiotherapy-induced changes in systemic immune parameters of head and neck cancer patients and proposed markers suitable for patient stratification worth investigating in larger patient cohorts.

## 1. Introduction

Head and neck cancers comprise of a group of malignancies arising in the upper aerodigestive tract (oral cavity, pharynx, larynx, paranasal sinuses and salivary glands) [1]. Based on recent cancer statistics released by the International Agency for Research on Cancer (IARC) the cumulative incidence of head and neck cancers was 887,659 new cases in 2018, which ranked these tumors as the seventh most common cancer types worldwide [2]. The incidence of head and neck cancer within European countries is very close to the worldwide average, being the fifth most common cancer in Hungary and France, the seventh in Poland, and the ninth in Germany and United Kingdom (based on data available in the Global Cancer Observatory database operated by IARC). The 5 years survival rate is under 65% due mostly to the late diagnosis in the absence of early symptoms [3]. About 90% of all head and neck cancers are squamous cell carcinomas (HNSCC) [4]. Apart of surgery the main treatment protocol for HNSCC is radiotherapy (RT) combined with chemotherapy [5].

The importance of an efficient anti-tumor immune response in achieving long-term recurrence- and metastasis-free tumor control has been recognized for a long time. The immune status of the patients is often significantly influenced by RT. Basically RT is considered a local therapeutic modality acting by direct energy deposition within the tumor bed leading to DNA lesions and extensive cell death. These effects are known in radiobiology as targeted effects. It has been recognized that local RT can act outside the directly irradiated site as well, leading to bystander responses (manifested in the vicinity to the directly irradiated tissue) and systemic responses (manifested systemically) [6]. These effects are called non-targeted effects. While the exact mechanism of RT-induced non-targeted effects is not yet elucidated, several candidate mediators of RT-induced bystander and systemic effects have been proposed, among others cell-based and soluble mediators of stress-, inflammatory- and immune response [7,8]. These RT-induced systemic changes interact with other antitumor treatment modalities, such as chemotherapy or, most importantly, immunotherapy and might influence long-term treatment outcome both in terms of local tumor control and development of distant metastases. Furthermore, these out-of-field effects are involved in the development of RT-induced normal tissue side effects by initiating inflammatory and immune reactions outside the directly irradiated tissues.

Therefore, investigation of RT-induced local and systemic changes in immune parameters could identify important predictive markers, which help in therapy individualization and identification of patient subgroups who benefit of certain therapeutic combinations. While several studies characterized baseline intratumoral immune parameters in different tumor types with the aim to identify long-term prognostic indicators [9,10,11,12,13,14,15,16,17], much less is known about how anticancer treatment interacts with local and systemic immune parameters and how these interactions impact therapy response and long-term prognosis.

The aim of the present work is to investigate the effect of RT on the systemic immune status by analyzing changes in the immune phenotype of peripheral blood mononuclear cells (PBMCs) and alterations in immune- and inflammation-related plasma proteins in HNSCC patients. We also determined radiotherapy-induced changes in the expression of previously identified radiation-responsive genes detectable systemically, in the peripheral lymphocytes. The current data indicate that local RT has a significant systemic effect, which lasts at least one month after completion of therapy.

## 2. Results

### 2.1. Clinical Parameters of HNSCC Patients

A group of 23 HNSCC patients (14 men and 9 women, aged 43–79 years) treated with IMRT were studied. Treatment protocols and clinical parameters including tumor type and localization, TNM stage, RT-induced acute mucosal reactions (AMR) and tumor response to RT are listed in Table 1. According to localization, 52% of patients had larynx tumours (*n* = 12), 35% of patients had oral cavity tumours (*n* = 8) and there were two cases with oropharynx tumour and one case with parotid tumour. All patients were human papilloma virus (HPV) negative. The p53 status was not evaluated.

Early treatment responses were evaluated one month after the completion of the radiotherapy. Treatment responses were very good; apart of one patient showing no tumor response, 2 patients with local recurrence and one patient with suspected recurrence the rest of the patients (*n* = 1 8, 78% of patients) had complete tumor response. 

Since the main criteria for patient selection was to receive RT only without chemotherapy, 91% of the selected patients had no lymph node involvement and none of the patients had distant metastasis at the time of diagnosis. However, the size of the primary tumor varied among patients (30% in T1 stage, 39% in T2 stage, 13% in T3 stage and 17% in T4 stage). Based on total dose received patients can be grouped in three categories (51–52.8 Gy, *n* = 7; 57.6–60 Gy, *n* = 7; 64.8–74 Gy, *n* = 9). A minor part of patients received high dose/fraction (3 Gy in 17 fractions) (*n* = 5), while the majority received 1.6–2.2 Gy/fraction in 30–40 fractions (*n* = 18). RT-related AMR were followed in patients and scored based on Radiation Therapy Oncology Group (RTOG) grading system according to the severity of the symptoms, with AMR0 indicating no mucosal side effects and AMR3 indicating serious mucosal ulcerations, inflammatory reactions and prolonged healing time. One patient had no AMR, 9% had AMR1, 39% AMR2 and 48% AMR3. Weak positive correlations were detected between the severity of AMR and total dose received (*r* = 0.2652) as well as between the severity of AMR and dose/fraction (*r* = 0.3368).

### 2.2. Radiotherapy Alters Gene Expression Profile of Peripheral Blood Cells

In order to determine RT-related gene expression changes in the peripheral blood cells of HNSCC patients, a set of 5 genes, which were previously reported to be radiation responsive were analyzed by qRT-PCR before start of RT, directly after and one month after the completion of RT. The 5 genes are: ferredoxin reductase (FDXR), sestrin 1 (SESN1), growth arrest and DNA damage inducible alpha (GADD45), damage specific DNA binding protein 2 (DDB2) and mouse double minute 2 (MDM2).

Alterations in the expression of all of the investigated genes were detected directly after the completion of RT, FDXR, GADD45, DDB2 and MDM2 expression increased, while SESN1 expression decreased compared to pre-treatment levels. Expression changes were moderate; the highest, 1.9-fold increase was noted for GADD45. One month after RT expression levels returned to pre-treatment values for FDXR and DDB2 but remained practically unchanged for the other 3 genes, indicating a persistent gene expression deregulation after RT (Figure 1). No statistically significant correlations were found between expression level of the analyzed genes and either total dose received or the severity of AMR.

### 2.3. Local Tumor Irradiation Induces Systemic Changes in the Level of Immune and Inflammation-Related Plasma Proteins

Changes in systemic inflammatory markers of HNSCC patients before and after RT were investigated next. First a high throughput protein array able to detect simultaneous changes in the plasma level of 105 proteins was used. The array was performed in a selected (*n* = 11) subgroup of patients and the aim was to screen for proteins with an altered secretion level. Since the used protein arrays yielded semi-quantitative data only, 8 proteins indicating radiation-related variations in plasma levels were further analyzed by enzyme-linked immunosorbent assay (ELISA) in all HNSCC patients (*n* = 23) and compared to levels in healthy controls (*n* = 6). These proteins were: adiponectin/Acrp30, apolipoprotein A-1 (ApoA1), B-cell activating factor (BAFF), CXC motif chemokine ligand 5 (CXCL5/ENA-78), cluster of differentiation 14 (CD14), trefoil factor 3 (TFF3), endoglin/CD105 and complement component C5/C5a.

Before RT both ApoA1 and endoglin concentrations were lower in the plasma of HNSCC patients than in healthy controls (1.6-fold and 1.3-fold decrease for ApoA1 and endoglin, respectively) suggesting disease-related changes. RT did not influence ApoA1 levels, which remained persistently decreased even one month after the completion of the therapy. In the case of endoglin RT induced a further, 1.4-fold decrease in its plasma concentration compared to pre-treatment values and this decrease persisted at one month after RT. Adiponectin and BAFF were unchanged in patients before therapy compared to healthy controls but showed significant increase (1.3-fold and 1.65-fold for adiponectin and BAFF respectively) one month after RT compared to pre-treatment values. CD14 showed similar tendency to adiponectin and BAFF but changes were milder. Quantitative changes in the other three investigated plasma proteins (complement component C5/C5a, CXCL5 and TFF3) were mild and statistically not significant (Figure 2).

### 2.4. RT Induces Changes in the Immune Phenotype of PBMCs of HNSCC Patients

Therapy-related changes in the cellular immune parameters of RT-treated HNSCC patients (*n* = 23) were investigated by immune phenotyping of the PBMCs before RT, directly after and one month after RT and compared to the immune phenotype of healthy controls (*n* = 18). We investigated the impact of RT on CD4 lymphocytes including Tregs as well as T cell activation markers, T cell proliferation status, natural killer (NK) cells, circulating dendritic cells (DCs) and myeloid-derived suppressor cells (MDSCs).

The fraction of CD4 cells within the lymphocyte population was not different in cancer patients before RT compared to healthy controls. Directly after RT the fraction of CD4 T cells decreased with 20% compared to pre-treatment values, while one month after RT this decrease became more pronounced (34%) indicating a strong radiation effect on the homeostasis of circulating CD4 pool (Figure 3A).

The Treg subpopulation within the CD4 cells was identified based on their expression of CD4 and Foxp3. The fraction of Treg cells in the PBMCs of healthy controls was low, accounting for 0.36% and showed a high inter-individual variation. The fraction of Treg cells in HNSCC patients increased 2-fold compared to healthy controls. RT led to a further progressive elevation in the proportion of Treg cells (Figure 3B). These data indicate that the tumorous state significantly increased the fraction of Treg cells in the peripheral blood, while RT accentuated this effect, which resulted in persistently elevated Treg levels for at least one month after the completion of RT.

Next we investigated markers reflecting the proliferation capacity, activation status and functional integrity of CD4 cells. CD4 proliferation was measured by the expression of the Ki67 proliferation marker. Proliferation of CD4 cells in HNSCC patients before treatment was not significantly different from that in healthy controls. However, RT induced a significant increase in the fraction of proliferating CD4 cells, leading to a 2-fold and 2.36-fold increase in the proportion of Ki67+ CD4 cells directly after and one month after RT respectively (Figure 4A). It is interesting to note that increased CD4 proliferation was present only in approx. 40% of patients and especially one month after RT patients could be separated in two well-delineated groups: one group with moderately increased Ki67 levels (1.4-fold increase in Ki67 levels compared to pre-treatment values) and one with strongly increased Ki67 levels (3.6-fold increase in Ki67 levels compared to pre-treatment values). The difference in the CD4 proliferation between the two subgroups was statistically significant (*p* = 0.0196).

The expression of CTLA4 marker on CD4 T-effector cells indicates the acquisition of an immune inhibitory phenotype. CTLA4 is constitutively present on most Treg cells at low levels; its upregulated expression indicates Treg activation. CTLA-4 levels on CD4+ T cells in healthy controls were low (2.66%). CTLA-4 expression on CD4 cells was increased with 190% in cancer patients before treatment compared to healthy controls, and RT induced a further strong and persistent increase in CTLA-4 expression (212% increase one month after RT compared to pre-treatment levels) (Figure 4B). Baseline and therapy-related CTLA4 expression was equally increased on both Treg cells and CD4+Foxp3- effector T (Teff) cells, indicating a shift of CD4 cells towards an immune suppressing phenotype (Appendix A). We analyzed relative CTLA4 expression one month after RT compared to pre-treatment levels in patients with moderate or strong CD4 proliferation. While CTLA4 expression one month after RT was higher in patients with highly proliferating CD4 cells compared to patients with moderately proliferating CD4 cells (2-fold increase versus 1.3-fold increase), differences were statistically not significant.

The PD-1 marker expressed on Teff cells indicates T cell anergy with altered capacity to react to CD4 activating stimuli. PD-1 expression on CD4 Teff cells showed a similar pattern to CTLA4 expression, having increased baseline levels and being further upregulated by RT (Figure 4C). Similarly to CTLA4, PD-1 expression was also higher one month after RT in patients with highly proliferating CD4 cells compared to patients with moderate CD4 proliferation (2.1-fold increase versus 1.4-fold increase) but these differences were statistically not significant. To note, that PD-1 was present on Treg cells as well at well-detectable levels, but neither malignant condition nor RT influenced its expression level.

The CD39 Treg marker identifies a certain Treg subpopulation with elevated potential to induce T-eff cell apoptosis. Interestingly, this Treg subpopulation was strongly decreased in cancer patients compared to healthy controls (approx. 4-fold decrease) but RT did not impact its expression (Figure 4D).

Circulating levels of NK cells (identified as CD16 CD56 double positive CD3- lymphocytes) were not affected either by the tumorous condition or by RT. Two subpopulations of circulating dendritic cells were evaluated: CD123+ plasmacytoid and CD11c+ myeloid DC. Both were strongly elevated in cancer patients before treatment (95-fold and 88-fold increase for the CD123+ and CD11c+ cells, respectively) compared to healthy controls. Although RT moderately reduced circulating CD123+ DC levels, overall DC levels remained strongly increased even one month after RT (Figure 5). This indicates that increased circulating DC levels were related to the malignant condition primarily and were only minimally influenced by RT itself.

The fraction of MDSCs present in the circulating blood was very low in healthy controls (below 1%), which in cancer patients before RT was 3-fold lower while RT did not influence MDSC levels. However, the fraction of CD14+ and CD14− MDSCs were largely different in pre-RT cancer patients and healthy controls. While CD14+ MDSCs represented approx. 35% of total MDSCs in healthy controls, in cancer patients before treatment approx. 70% of total MDSCs were CD14+ (Figure 6).

The influence of RT on polymorphonuclear cell (PMC) and B cell level was investigated in fresh whole blood of a limited number of patients (*n* = 7). One aim of this experiment was to test the feasibility of whole blood collection and shipping without compromising blood cell parameters. In these experiments we compared the level of neutrophils, basophils and eosinophils as well as B cells in HNSCC patients before and one month after RT. No significant changes were seen in the number of PMCs, while B cell numbers decreased significantly one month after RT compared to pre-treatment levels (Appendix A).

Summarizing phenotypical changes in PBMCs (Table 2), the tumorous condition had a significant impact on the pool and functional integrity of most of the analyzed PBMC subpopulations. RT either did not modify pre-treatment values or it further accentuated pre-treatment changes. The only exception was noticed in the case of CD123+ DCs, where RT mildly reduced elevated pre-treatment DC levels. Those five patients, who responded poor to treatment (see Table 1) did not display any specific response pattern in terms of gene expression or immunological alterations. Seven out of the eight patients with strong CD4 proliferation one month after radiotherapy were in T2–T4 stage based on the TNM tumor staging at the time of diagnosis.

## 3. Discussion

A major problem in planning individualized anticancer treatment and monitoring therapy response for most cancer types is the lack of suitable biomarkers indicating minimal residual disease, predicting tumor relapse and development of distant metastasis, allowing patient stratification for optimal response rate to a certain therapy and identifying those patients, who are at increased risk for developing therapy-related side effects. Given the high heterogeneity and molecular diversity of head and neck tumors, a large panel of molecular mediators of tumor progression, invasion and metastasis have been described, but most of them await further biological interpretation and validation. Around 80% of HNSCC cases present downregulation of p53, either via TP53 mutation (more than 50% of HPV negative HNSCC cases) or other pathways like MDM2 or cyclin dependent kinase inhibitor 1A (CDKN1A) or CDKN2A (p16), but changes in cyclin dependent kinases (CDKs), CUB and Sushi multiple domains (CSMD), nuclear factor κB (NF-κB), phosphatidylinositol-4,5-bisphosphate 3-kinase, catalytic subunit alpha (PIK3CA), transforming growth factor beta (TGFβ), vascular endothelial growth factor (VEGF) genes are also described [18]. Some of these molecules have been proposed as prognostic markers in different studies [19,20,21,22]. The presence of circulating tumor DNA or circulating HPV DNA was also investigated in multiple studies for their prognostic or predictive value [23,24,25].

Several studies were conducted to determine radiation-induced gene expression changes in healthy cells, which could serve as markers of normal tissue radiation sensitivity. GADD45A, SESN1, FDXR, DDB2 and MDM2 were described as radiation responsive genes and were proposed as candidates to predict individual response to radiation [26,27,28,29,30]. These gene expression changes can manifest on directly irradiated cells (fibroblasts or PBMCs) and were evaluated shortly after irradiation. Our data show that blood cells of RT treated HNSCC patients carried similar gene expression changes as reported by the above cited studies. Even if changes in gene expression levels were milder, it is very important to highlight that these expression profiles come from blood samples from partially irradiated patients and they were present even one month after the completion of RT, indicating persistent transcriptional deregulation. Whether these persistent gene expression changes present at systemic level can be correlated with an increased individual radiosensitivity, needs further studies in patients followed for longer time.

Soluble proteins can also serve as important markers indicating therapy response and potentially predicting therapy-related side effects. There have been several attempts to identify plasma markers for clinical outcome, prognosis or therapy response without a consensus about the most promising prognostic biomarkers, or whether there is a unique biomarker protein or they should be combined into profiles. In a multiplex bead-based approach, Brondum et. al. analyzed 19 previously described proteins in head and neck cancer patients before treatment, and they found that there was a significant difference between the baseline levels of the patient and the control group in IL-2, IL-4, EGFR, OPN, VEGFR-1, VEGFR-2, VEGF and GRO levels, but none of these markers could be correlated with outcome [31]. Widlak and his co-workers investigated changes in the proteome of head and neck cancer patients before and at various time points after irradiation and identified 55 proteins up- or downregulated post-RT compared to pre-RT values. However, most were short-term changes and only approximately 20 protein changes persisted one month after RT. Functionally, these were related to inflammation, acute response, innate immunity and lipid metabolism [32]. Using a protein array consisting of 105 immune and inflammation-related plasma proteins we screened for quantitative changes in HNSCC patients before and after RT and identified 8 proteins with altered secretion levels. While the number is moderately lower than reported by Widlak et. al. functionally the majority of the altered proteins were similar, being involved in inflammation, immune response or lipid metabolism. To note that approx. half of the proteins, were also significantly different in pre-RT patients compared to healthy controls, with changes persisting or increasing by one month after RT, indicating persistent deregulation in blood proteome. Similarly to Widlak et. al. we could not correlate protein alterations with the severity of acute mucosal reactions. Though, the numbers of patients in both studies were not big enough to draw significant conclusions on correlations between AMR and protein changes. In our study endoglin was one of the proteins affected by RT in HNSCC patients. Functionally endoglin is involved in angiogenesis and intratumoral endoglin level is an important marker of tumor angiogenesis and neovascularization process [33,34]. Both intratumoral endoglin expression and circulating endoglin levels were correlated with therapy response and long-term prognosis [35,36]. Li et. al. showed that circulating endoglin levels could ameliorate RT-induced late-fibrosis in breast cancer patients by forming stable complex with TGF-β1 and thus diminish the effect of TGF-β1 in inducing tissue fibrosis. In their study an inverse correlation was found between circulating endoglin levels and the severity of fibrosis [37]. This suggests endoglin could be an important marker of late radiation toxicity in head and neck cancer patients as well. ApoA1 was another protein, which decreased significantly in pre-RT patients in our study and remained decreased post-RT as well. This is in line with previous reports demonstrating strong correlation between decreased circulating ApoA1 levels and the risk of developing distant metastasis as well as overall prognosis [38,39].

In HNSCC the amount of certain immune cell populations and expression levels of various activation and suppressive markers on their cell surface might be altered. Most studies agree that increased Treg infiltration (specifically defined Treg subpopulations expressing various activation markers such as CTLA4, PD-1 or Helios) or increased infiltration of CD4 or CD8 T cells expressing PD-1 is a negative prognostic marker [40,41] in HNSCC, while the presence of memory CD103+ T cells or Tbet+ Tregs correlate with a better prognosis [42,43]. Circulating Treg cells were correlated with negative clinicopathological findings and worse prognosis [44,45,46], on the other hand increased levels of circulating Th17 cells were considered a good prognostic indicator [47].

Changes in local and systemic immune responses also reflect patients’ responses to RT, therefore immune-related markers harbor a great potential in becoming key indicators for therapy individualization and prediction of treatment response. In a large German multicenter study it was concluded that CD8+ tumor infiltrating lymphocytes constitute an independent prognostic marker in HNSCC patients treated with adjuvant chemoradiotherapy and can potentially be used for patient stratification [48]. Stably increased levels of circulating Tregs present in the blood of head and neck cancer patients treated with chemoradiotherapy have been correlated with recurrence [49].

Our data show that baseline systemic CD4 levels were similar in HNSCC patients to healthy controls, while the fraction of Treg cells was almost two times higher. Both the fraction of CTLA4-expressing CD4 cells and PD-1 expressing Teff cells were significantly higher in patients before treatment than in healthy controls. These data support the existence of a systemic immune suppression in HNSCC patients before the start of RT as already reported [50,51]. CD4 cells and notably Treg cells were the major PBMC subpopulations affected by RT in HNSCC patients. CD4 cells progressively decreased after RT, while Treg cells progressively increased. Since the fraction of proliferating CD4 cells increased after RT, most probably Treg cells proliferated better than Teff cells, leading to a shift in the CD4/Treg ratio in the favor of Treg cells as also shown before [52,53]. RT-induced increase in the fraction of CTLA4-expressing Teff and Treg cells as well as increased PD-1 expression on Teff cells indicate that systemic immune suppression became more severe as a result of RT. Previously we found that irradiation increased CTLA4 expression on circulating CD4 cells in healthy mice and Muroyama et. al. reported similar findings in the microenvironment of locally irradiated murine tumors [52,53]. It has also been shown that RT combined with chemotherapy increased PD-1 expression of T cells [54,55]. There was a tendency that patients with high CD4 proliferation kinetics one month after RT had increased CTLA4 and PD-1 levels on CD4 effector cells. However, due to the small number of patients in each subgroup data were statistically not significant. It was also interesting the finding that 87% of patients with highly proliferating CD4 cells and increased CTLA4 and PD-1 expression had tumors in T2-T4 stage. We think this indicates that a more advanced initial tumor stage leads to a stronger and more persistent systemic immune suppression after RT. Though, in order to improve the statistical power of these data, larger patient groups need to be investigated.

MDSCs are immature myeloid cells released from the bone marrow in the peripheral blood having immune suppressive properties. Increased MDSC levels have been described in several tumor types including HNSCC and have been considered as independent prognostic factors of poor outcome [56,57,58,59]. Chemoradiotherapy was reported to increase total and CD14+ MDSC levels in HNSCC patients [55,60]. In contrast to previous reports we found decreased baseline MDSC levels (identified as HLA-DR-CD11+CD33+ cells) compared to healthy controls though the fraction of CD14+ MDSCs was increased. RT induced a progressive normalization in MDSC levels.

Both increased [61] and decreased [62,63] circulating NK cell levels were reported in HNSCC patients. In the current study we could not detect any changes in circulating NK cell levels either before or after RT. Several studies have investigated circulating DC levels and their relation with other circulating immune cells [64,65] and have been correlated with disease prognosis [66]. We showed strong increase in the fraction of circulating DCs in HNSCC patients before RT affecting both myeloid and plasmacytoid DCs, which was only slightly modified by the therapy. Very similar findings were reported by Schmidt et. al. who investigated circulating levels of myeloid and plasmacytoid DCs in breast and prostate cancer patients before and after radiochemotherapy [64]. In contrast other publications reported decreased baseline circulating DC levels with an immature phenotype in HNSCC patients [65]. However, it has also been shown that DCs are often functionally deficient in cancer patients, thus, circulating DC levels do not necessarily reflect their functional integrity [67].

Based on these findings we think that immunosuppression in the HNSCC patients was realized by two main mechanisms: first, through increased CTLA4 expression on both Treg and Teff CD4 cells the CD4-DC interactions were compromised, turning DCs tolerogenic. Second, due to the elevated PD-1 levels Teff cells became anergic, impairing their capacity to react to immune activating stimuli or induce immune activation. Interestingly, it seems that MDSCs did not participate in this immune suppression, though we cannot exclude the role of the relative increase in the fraction of CD14+ cells within the MDSCs. We showed that RT further impaired systemic immune suppression. It is important to note that immunosuppression after RT was a relatively persistent process since the above mentioned immune parameters did not show any tendency for normalization one month after the completion of RT. This information is important, since currently a major research area in oncology is to optimally combine RT with various immunotherapeutic approaches to reach synergistic antitumor effects. In order to find the best combination a detailed knowledge is needed on how RT modulates immune parameters within the tumor and systemically. There are studies indicating that an immunogenic phenotype of the tumor before RT is needed for radiation to increase tumor immunogenicity and RT can only amplify an immunogenic phenotype but cannot change a net immune suppressing microenvironment into an immunogenic one [7,68]. At present RT combined with immune checkpoint blockade therapy is a promising combination. The exact mechanism how this combination achieves synergy is not yet elucidated and also the optimal timing of RT and immunotherapy is not yet fixed [69,70,71]. Our findings suggest that immune checkpoint blockade applied first would serve to eliminate the immune suppressing T cell environment, while RT applied subsequently, would increase the release of immunogenic tumor antigens, which could initiate an efficient antitumor immune response.

## 4. Materials and Methods

### 4.1. Sample Collection from Radiotherapy Patients

Twenty-three patients (9 women and 14 men) aged from 43 to 79 years (median 64 years) treated with intensity modulated radiotherapy (IMRT) for HNSCC, who had no previous surgery or chemotherapy, were recruited for the study; patients were treated at the Maria Skłodowska-Curie Institute—Oncology Center, Gliwice Branch, Poland, between January and September 2016. Different RT schemes with a dose/fraction ranging from 1.6 to 3.0 were implemented; total radiation doses delivered to gross tumor volume were between 51 and 74 Gy (median 57.6 Gy) and overall treatment time was in the 23–61 days range (median 44 days). Demographics and clinical information as well as details of RT are presented in Table 1. Blood samples were collected within one week before the start of treatment, directly after the last fraction (within 6 h), and approximately one month after completion of the therapy. Control blood samples were collected from age-matched healthy volunteers at the National Institute of Oncology, Budapest, Hungary.

### 4.2. Ethical Permissions

The study was conducted in accordance with the Declaration of Helsinki, and the protocol was approved by the Ethics Committee of the Bioethical Committee in Maria Sklodowska-Curie Institute, Warsaw, approval number 27/2015 from 18/08/2015 as well as Scientific and Research Ethics Committee of the Hungarian Medical Research Council (16738-2/2015/EKU). All subjects gave their informed consent for inclusion before they participated in the study.

### 4.3. Isolation of PBMCs from Human Blood

Approximately 15 mL human blood/patient/sampling time was collected in heparinised tubes (Vacuette tube, Greiner Bio-One Gmb, 4550 Kremsmünster, Austria) and diluted with phosphate buffered saline (PBS) in 1:1 ratio. 6 mL of diluted blood was pipetted on 2 mL Histopaque (Sigma-Aldrich, Co., 3050 St. Louis, MO, USA) and centrifuged (Thermo Scientific, Heraeus Megafuge 16R Centrifuge) for 15 min at 2000 rpm at room temperature in zero fall-off mode. PBMCs were then isolated from the corresponding layer which is under the blood plasma layer and above Histopaque layer. Cells were washed twice in PBS. PBMCs were counted and aliquoted in RPMI-1640 containing 10% dimethyl sulfoxide and 50% fetal bovine serum at 1.5 × 10^6^ cells/mL, gradually cooled down to −80 °C using a NALGENE™ Cryo 1 °C Freezing Container device filled with 2-propanol puriss (Reanal Labor, Budapest, Hungary) for 24 h in CryoPure tubes (Sarstedt, Nümbrecht, Germany) and stored in liquid nitrogen until assayed or shipped to Hungary in dry ice for PBMC phenotyping.

The supernatant (blood plasma) was centrifuged again (10 min, 2500 rpm, 25 °C) to remove residual cellular debris. Plasma samples were stored at −70 °C until assayed or shipped to Hungary in dry ice for protein expression analyses. All samples shipped to Hungary were processed within 6 months.

From a limited number of patients (*n* = 7) whole blood was collected, and shipped to Erlangen, Germany at 4 °C, within 24 h after collection, where it was processed immediately.

### 4.4. Immune Phenotyping of PBMCs and Polymorphonuclear Cells

Frozen PBMC samples were thawed in a 37 °C water bath and the number of viable cells was determined using trypan blue staining. One million cells were suspended in 100 µL staining buffer (consisting of Hank’s balanced salt solution containing 1% bovine serum albumin) in FACS tubes (Sarstedt, Nümbrecht, Germany). For cell surface staining cells were incubated with the corresponding antibodies at 4 °C in dark for 30 min. For intracellular staining cells were fixed and permeabilized using Biolegend FoxP3 Fix/Perm Buffer set (BioLegend, San Diego, CA, USA) according to the manufacturer’s instructions, followed by the addition of the corresponding antibodies. Stained cells were stored in 1% paraformaldehyde until assayed. The following directly labelled anti-human monoclonal antibodies were used for phenotypical analysis: CD4-APC, CTLA-4-PE, PD-1-PerCP/Cy5.5, FoxP3-Alexa Fluor488, CD39-PerCP/Cy5.5 and Ki-67-PE for Tregs; CD11b-FITC, CD14-PE, CD33-PerCP/Cy5.5 and HLA-DR-APC for MDSCs; CD16-FITC, CD56-PE and CD3-APC for NKs and Lineage coctail1 (CD3, CD14, CD16, CD19, CD20, CD56)-FITC, HLA-DR-PerCP, CD123-PE and CD11c-APC for DCs. All antibodies were purchased from BioLegend.

For the immune phenotyping of polymorphonuclear cells fresh whole blood samples were processed within 24 h after collection and analyzed by an immunophenotyping blood (DIoB) assay [72].

### 4.5. Hierarchical Gating Strategy

PBMC subpopulations were analysed and quantified by flow cytometry (FACS Calibur, Becton Dickinson, CA, USA). A hierarchical gating strategy was created for effective flow cytometry data analysis using the Kaluza 1.5a software. CD4 cells (including Tregs) and NK cells were analyzed within the lymphocyte gate. Treg cells were quantified as Foxp3+ CD4+ cells. CTLA4+ cell subpopulations were evaluated within the whole CD4+ gate as well as within the individual Treg gate (CD4+Foxp3+) and Teff gate (CD4+Foxp3-). PD1+ cells were evaluated within the CD4+ gate in a similar manner to CTLA4. CD39+ cells were evaluated only in the Treg gate (Appendix A). Proliferating index of CD4 cells was determined evaluating the fraction of Ki67+ cells, applying the same gating strategy as for CTLA4 and PD1. NK cells were identified as CD3- cells within the lymphocyte gate showing CD16+ and CD56+ double positivity (Appendix A). DCs were analysed within all PBMCs. Myeloid DCs were considered CD11c+ Lineage-1- cells, while lymphoid or plasmacytoid DCs were considered CD123+ Lineage-1- cells (Appendix A). MDSCs were identified in the total PBMCs excluding the lymphocyte gate as CD11b+ and CD33+ double positive cells within the HLA-DR- cells. The fraction of CD14+ and CD14− cells was evaluated within the total HLA-DR-CD11b+CD33+ MDSCs (Appendix A). Neutrophils were quantified as SSC^hi^ CD16+ cells, eosinophils as SSC^hi^ CD16- and the definition of basophils was CD3-/CD14-/(CD16-)/CD19-/CD20-/CD56-/HLA-DR-/CD123+ as previously published [72].

### 4.6. Analysis of Plasma Proteins in HNSCC Patients Treated with RT

The concentration of 105 inflammation-related plasma proteins was quantified in 11 head and neck cancer patients before and after RT by Proteome Profiler Antibody Arrays (Human XL Cytokine Array Kit, R&D Systems, Minneapolis, MN, USA). 350 µL plasma samples were loaded on each array membrane, followed by incubations with biotinylated detection antibody cocktails and with streptavidin-HRP. Results were visualized with chemiluminescence detection reagents and exposed on X-ray films (CL-XPosure Film, Thermo Scientific, Rockford, United States). Evaluation was performed with ImageJ software.

Proteins showing statistically significant changes on the protein array were analyzed by ELISA. The following ELISA reactions were performed according to the manufacturer’s instructions: BAFF, adiponectin/Acrp30, CXCL5/ENA-78, ApoA1, CD14, TFF3, endoglin/CD105 and complement component C5/C5a. All ELISA kits were purchased from R&D Systems (Minneapolis, MN, USA).

### 4.7. RNA Isolation and Reverse Transcription

Total RNA was extracted with the PAXgene Blood miRNA kit (Qiagen, PreAnalytiX GmbH, Hilden, Germany) from blood samples collected in PAXgene tubes from the same RT patients as enrolled in immune phenotypical analysis using a robotic workstation Qiacube (Qiagen, Manchester, UK). The quantity of isolated RNA was determined by spectrophotometry with a ND-1000 NanoDrop and quality was assessed using a Tapestation 220 (Agilent Technologies, CA, USA). cDNA was prepared from 350 ng of the total RNA using High Capacity cDNA reverse transcription kit (Applied Biosystems, FosterCity, CA, USA) according to the manufacturer’s protocol.

### 4.8. Quantitative Real-Time Polymerase Chain Reaction

QRT-PCR was performed using a Rotor-Gene Q (Qiagen, Hilden, Germany) with PerfeCTa MultiPlex qPCR SuperMix (Quanta Bioscience, Inc., Gaithersburg, MD, USA). The samples were run in triplicates in 10 µL reactions with 1 µL of the cDNA synthesis reaction together with six different sets of primers and fluorescent probes at 300 nM concentration each. 3′6-Carboxyfluorescein (FAM), 6-Hexachlorofluorescein (HEX), Atto 680, Atto 390, Texas Red (Eurogentec Ltd., Fawley, Hampshire, UK) and CY5 (Sigma-Aldrich, Poole, Dorset, UK) were used as fluorochrome reporters for the probes analyzed in multiplexed reactions with 6 genes per run including a housekeeping gene. The investigated genes were the following: hypoxanthine phosphoribosyltransferase 1 (HPRT1) as housekeeping gene, DDB2, GADD45, SESN1, FDXR and MDM2. Primer sequences used for the amplification are shown in Table 3. The reactions were performed with the following cycling conditions: 2 min at 95 °C, then 45 cycles of 10 s at 95 °C and 60 s at 60 °C. Data were collected and analyzed by Rotor-Gene Q Series Software. Gene target Ct (cycle threshold) values were normalized to HPRT1 internal control. Ct values were converted to transcript quantity using standard curves obtained by serial dilution of PCR-amplified DNA fragments of each gene. The linear dynamic range of the standard curves covering six orders of magnitude (serial dilution from 3.2 × 10^−4^ to 8.2 × 10^−10^) gave PCR efficiencies between 93 and 103% for each gene with *R^2^* > 0.998.

### 4.9. Statistical Analyses

Statistical analyses were performed with the GraphPad Prism version 6.00 for Windows software (GraphPad Software, La Jolla, CA, USA). All graphs are presented as mean  ±  standard deviation (SD). Two-tailed unpaired *t*-test was used to test differences between healthy controls and cancer patients and paired *t*-test to test differences among cancer patients before and after RT. *p*-values lower than 0.05 were considered statistically significant. Correlation analyses were executed with Pearson and Spearman correlations.

## 5. Conclusions

In conclusion, we demonstrated that HNSCC patients exhibit a systemic immune suppression caused by the malignant condition, which manifested in a compromised CD4 cell function and increase in the fraction of immune suppressive Tregs and effector CD4 cells. A major finding of our study was that RT further accentuated this immune suppressive phenotype, which persisted minimum one month after RT. By this we provided insight into how RT shapes systemic antitumor immune response, which could contribute to the optimization of combined immunotherapeutic and radiotherapeutic approaches and help in patient stratification to find those patient categories that could benefit from combined treatments. Furthermore, our study identified RT-induced systemic changes in gene expression and plasma protein secretion. We showed that genes identified as radiation responsive in directly irradiated tissues in previous studies had persistently altered gene expression pattern in the peripheral blood of RT-treated patients. Thus, we identified a panel of immune and inflammation-related markers detectable in the peripheral blood, which could help in a better patient stratification, in order to choose the optimal therapy combination for each individual patient. Since the relatively low patient number is a limitation of our study further prospective studies with much larger patient cohorts followed in the long run are needed in order to validate the predictive value of the proposed markers.

## Figures and Tables

**Figure 1 cancers-11-01324-f001:**
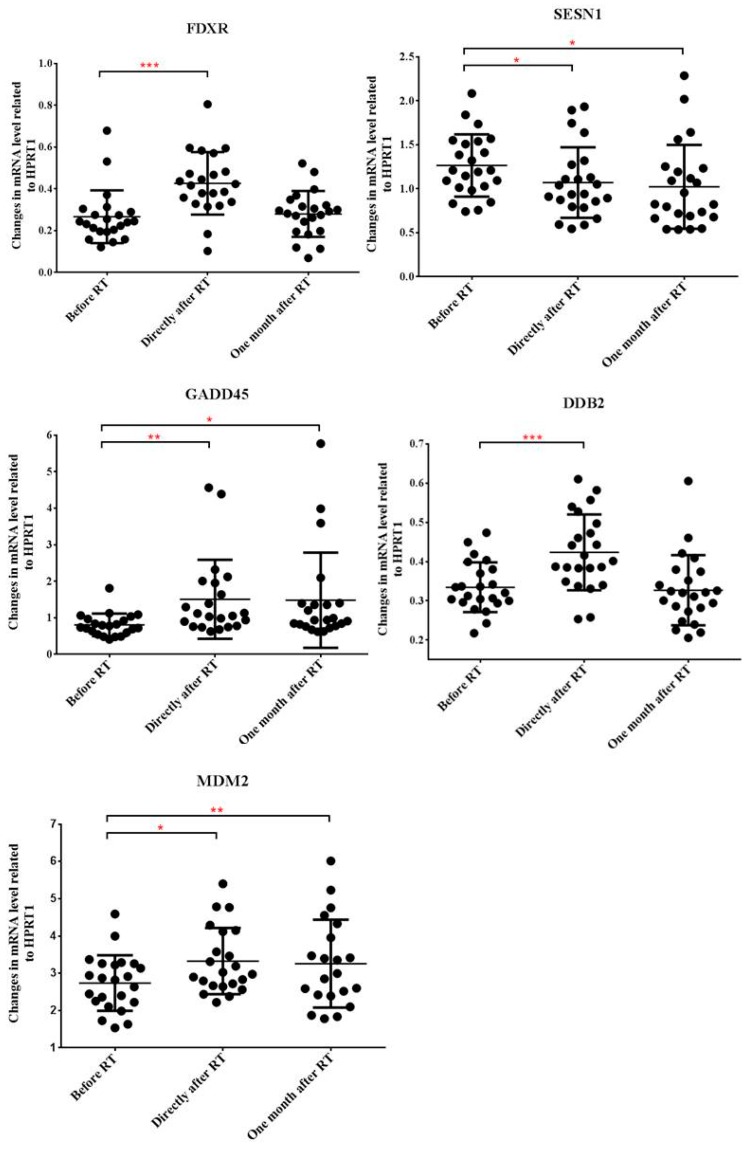
Expression of FXDR, SESN1, GADD45, DDB2 and MDM2 genes in the peripheral blood cells of head and neck squamous cell carcinoma (HNSCC) patients treated with RT. Blood samples were collected before the start of the treatment, directly after and one month after the last fraction of RT. RNA was isolated and qRT-PCR was performed as described in Materials and methods. Relative changes in the gene expression levels were compared between the three time points. Individual data points are shown together with the mean ± SD (*n* = 23). Statistical analyses were performed in log transformed data. Significant differences (Paired-T-test, *p* ≤ 0.05) are indicated (* *p* < 0.05, ** *p* < 0.01, *** *p* < 0.001).

**Figure 2 cancers-11-01324-f002:**
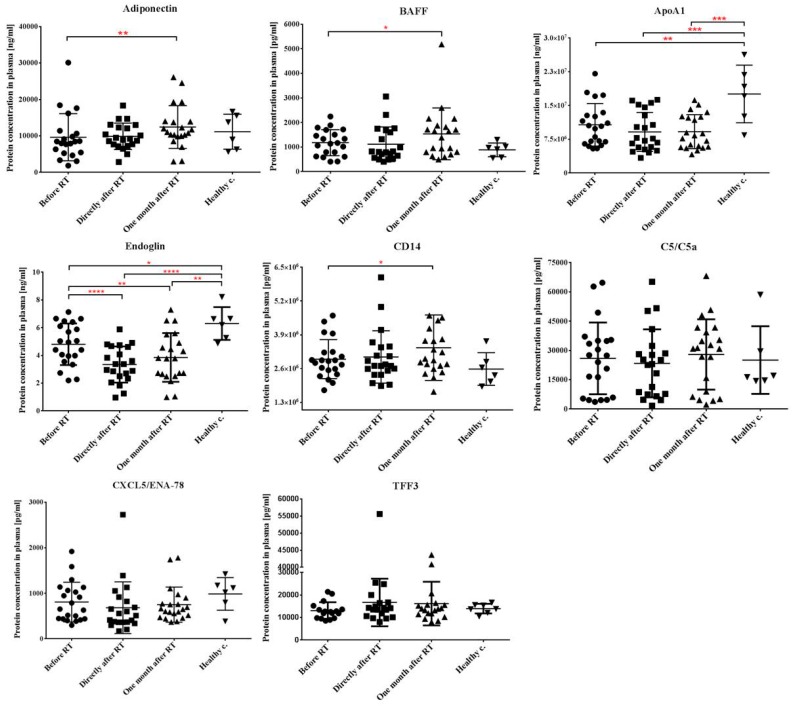
Levels of immune- and inflammation-related proteins in the plasma of HNSCC patients before and after radiotherapy. Blood was collected from cancer patients before the start of the treatment, directly after and one month after the last fraction of RT as well as from healthy volunteers. Plasma was isolated and concentration of the indicated proteins was measured by ELISA as described in Materials and methods. Absolute levels of protein concentration are indicated. Data are shown as individual data points (*n* = 23 for patients, *n* = 6 for healthy controls) together with the mean ± SD. Significant differences were tested with paired-*t*-test with 95% confidence interval (* *p* < 0.05, ** *p* < 0.01, *** *p* < 0.001, **** *p* < 0.0001).

**Figure 3 cancers-11-01324-f003:**
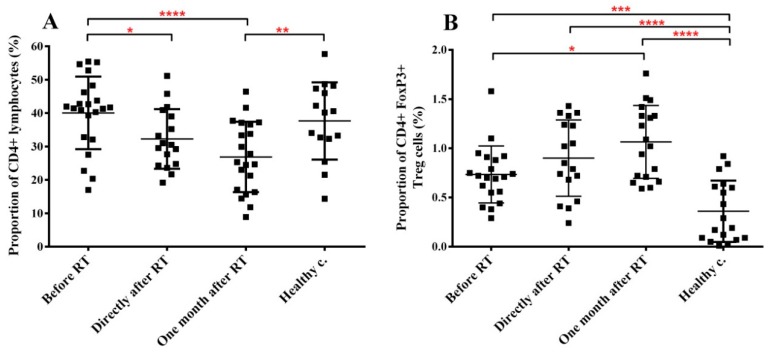
Radiotherapy reduces the total CD4+ T cell pool but increases the Treg pool within the lymphocyte population of H&N cancer patients. PBMCs were isolated from RT-treated H&N cancer patients, processed and labelled with the corresponding antibodies as described in Materials and methods s. (**A**) fraction of CD4+ cells within the lymphocyte gate. (**B**) fraction of CD4/Foxp3 double positive cells within the lymphocyte gate. Student’s paired *t*-test was used to test significant differences among the three time points of cancer patients’ samples, while unpaired *t*-test was used to test significant differences between the three time points and healthy control group using 95% confidence interval. Data are shown as individual data points together with the mean ± SD (cancer patients *n* = 23, healthy controls *n* = 15). Significant differences were indicated with * (*p* < 0.05), ** (*p* < 0.01), *** (*p* < 0.001) and **** (*p* < 0.0001).

**Figure 4 cancers-11-01324-f004:**
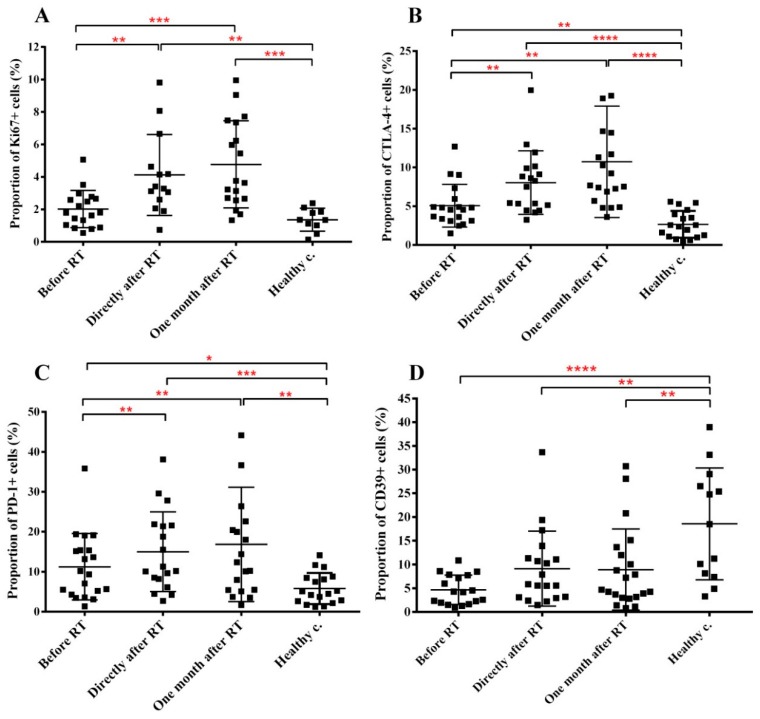
Tumorous condition and RT alters the proliferation and functional integrity of CD4+ cells. PBMCs were isolated from RT-treated H&N cancer patients, processed and labelled with the corresponding antibodies as described in Materials and methods. (**A**) fraction of Ki67+ CD4 cells within the CD4 cell pool. (**B**) fraction of CTLA4 expressing CD4 cells within the CD4 cell pool. (**C**) fraction of PD-1 expressing effector CD4 cells within the total effector CD4 cell population. (**D**) fraction of CD39+ Treg cells within the total Treg population. Student’s paired *t*-test was used to test significant differences among the three time points of cancer patients’ samples, while unpaired *t*-test was used to test significant differences between the three time points and healthy control group using 95% confidence interval. Data are shown as individual data points together with the mean ± SD (cancer patients *n* = 23, healthy controls *n* = 15). Significant differences were indicated with * (*p* < 0.05), ** (*p* < 0.01), *** (*p* < 0.001) and **** (*p* < 0.0001).

**Figure 5 cancers-11-01324-f005:**
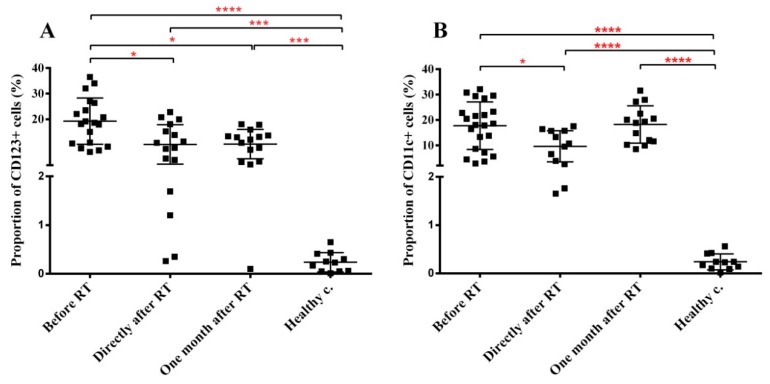
The fraction of circulating dendritic cells increases in the peripheral blood of H&N cancer patients and this increase is maintained after RT, as well. PBMCs were isolated from RT-treated H&N cancer patients, processed and labelled with the corresponding antibodies as described in Materials and methods. (**A**) fraction of CD123+ DCs. (**B**) fraction of CD11c+ DCs. Student’s paired *t*-test was used to test significant differences among the three time points of cancer patients’ samples, while unpaired *t*-test was used to test significant differences between the three time points and healthy control group using 95% confidence interval. Data are shown as individual data points together with the mean ± SD (cancer patients *n* = 23, healthy controls *n* = 15). Significant differences were indicated with * (*p* < 0.05), ** (*p* < 0.01), *** (*p* < 0.001) and **** (*p* < 0.0001).

**Figure 6 cancers-11-01324-f006:**
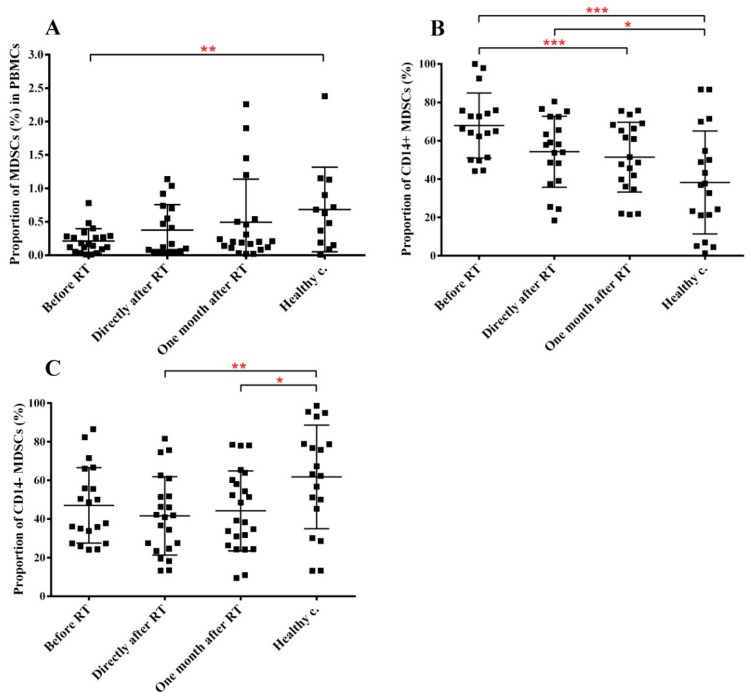
The fraction of MDSCs decreases in the peripheral blood of H&N cancer patients and this decrease is maintained after RT, as well. PBMCs were isolated from RT-treated H&N cancer patients, processed and labelled with the corresponding antibodies for identifying MDSCs as described in the Materials and methods section. We investigated two subpopulations of MDSCs. (**A**) fraction of MDSCs in PBMCs. (**B**) fraction of CD14+ subpopulation within all HLA-DR- cells. (**C**): fraction of CD14− subpopulation within all HLA-DR- cells. Student’s paired *t*-test was used to test significant differences among the three time points of cancer patients’ samples, while unpaired *t*-test was used to test significant differences between the three time points and healthy control group using 95% confidence interval. Data are shown as individual data points together with the mean ± SD (cancer patients *n* = 23, healthy controls *n* = 15). Significant differences were indicated with * (*p* < 0.05), ** (*p* < 0.01), *** (*p* < 0.001) and **** (*p* < 0.0001).

**Table 1 cancers-11-01324-t001:** Clinical parameters of head and neck squamous cell carcinoma patients

Patient Code	Pathology	Localization	TNM Classification	Gender	Age	Total Dose (Gy)	Number of Fractions	Dose/ Fraction (Gy)	Overall Treatment Time (days)	AMR (Acute Mucosal Reaction)	Response to Treatment
1_HNC	Carcinoma planoepitheliale G2	larynx (glottis)	T1N0M0	man	62	51	17	3	25	3	1
2_HNC	Carcinoma planoepitheliale akeratodes	larynx	T2N0M0	man	72	64.8	37	1.8	38	2	1
3_HNC	Carcinoma planoepitheliale keratodes G2	oral cavity	T2N0M0	man	44	52.8	33	1.6	61	3	4
4_HNC	Carcinoma planoepitheliale	larynx	T1N0M0	man	66	51	17	3	26	3	1
10_HNC	Squamous cell carcinoma	oral cavity	T1N0M0	man	43	52.8	33	1.6	48	2	1
12_HNC	Carcinoma planoepiheliale keratodes	larynx	T1N0M0	man	76	51	17	3	25	3	1
14_HNC	Carcinoma planoepitheliale keratodes	oral cavity	T4N0M0	man	63	57.6	36	1.6	50	1	6
17_HNC	Carcinoma planoepitheliale keratodes G1	larynx	T2N0M0	man	64	72	40	1.8	38	3	1
18_HNC	Carcinoma planoepitheliale invasium G1 keratodes	larynx	T1N0M0	man	70	51	17	3	23	2	1
19_HNC	Carcinoma planoepitheliale	larynx (epiglottis)	T2N0M0	woman	67	72	40	1.8	41	3	1
20_HNC	Carcinoma planoepitheliale G3	oral cavity	T4N0M0	man	59	57.6	36	1.6	50	0	1
21_HNC	Carcinoma planoepitheliale keratodes G1	larynx (epiglottis)	T3N2M0	woman	52	74	40	1.8	41	3	1
23_HNC	Carcinoma planoepitheliale non keratodes G2	larynx	T2N0M0	woman	57	66	30	2.2	48	3	3
24_HNC	Carcinoma planoepitheliale akeratodes	larynx (glottis)	T1N0M0	man	55	51	17	3	24	1	5
25_HNC	Carcinoma planoepitheliale keratodes G1	oral cavity (tonque)	T2N0M0	man	66	57.6	36	1.6	55	2	1
26_HNC	Carcinoma planoepitheliale keratodes	oropharynx (tonsil)	T3N0M0	man	73	70	35	2	51	2	1
27_HNC	Carcinoma planoepitheliale G1	oral cavity	T4N0M0	woman	49	57.6	36	1.6	50	2	1
28_HNC	Carcinoma planoepitheliale in situ	larynx	T2N0M0	woman	65	70.2	39	1.8	40	3	1
30_HNC	Carcinoma planoepitheliale akeratodes G2	larynx	T3N0M0	woman	68	72	40	1.8	41	3	1
31_HNC	Carcinoma planoepitheliale keratodes G1	oral cavity	T4N0M0	man	67	57.6	36	1.6	50	2	4
32_HNC	Carcinoma planoepitheliale keratodes G1	oropharynx	T2N0M0	woman	79	57.6	36	1.6	52	2	1
33_HNC	Low Grade Mucoepidermoid Carcinoma	parotid	T1N0M0	woman	57	66	33	2	46	3	1
34_HNC	Squamous cell carcinoma	oral cavity (tonque)	T2N2M0	woman	60	60	30	2	44	2	1

The scale of individual response to treatment is from 0 to 6 (0 = lost from control, 1 = complete tumor response, 2 = partial tumor response, 3 = no tumor response, 4 = local recurrence, 5 = lymph node recurrence, 6 = suspected recurrence). TNM staging system was used to determine the extent of cancer (T: size of primary tumor, N: degree of spread to regional lymph nodes, M: distant metastasis). Acute mucosal reaction (AMR) scores are based on Radiation Therapy Oncology Group (RTOG) grading. Grades 0–1 means minimum reactions and grades 2–3 indicate serious mucosal ulcerations.

**Table 2 cancers-11-01324-t002:** Summary of phenotypical changes in specific PBMC subpopulations in H&N cancer patients before and after RT compared to healthy controls.

Cell Population	Before the Radiotherapy	Directly after Radiotherapy	1 Month after Radiotherapy
CD4+ lymphocytes	n.s.	n.s.	↓ **
CD4+ FoxP3+ Treg	↑ ***	↑ ****	↑ ****
CTLA-4+ in CD4+ lymphocytes	↑ **	↑ ****	↑ ****
PD-1+ in CD4+ FoxP3- lymphocytes	↑ *	↑ ***	↑ **
CD39+ in CD4+ FoxP3+ lymphocytes	↓ ****	↓ **	↓ **
Ki-67+ in CD4+ lymphocytes	n.s.	↑ **	↑ ***
MDSCs in PBMCs	↓ **	n.s.	n.s.
CD14+ in MDSCs	↑ ***	↑ *	↑
CD14− in MDSCs	↓ *p* = 0.0569	↓ **	↓ *
CD123+ DCs	↑ ****	↑ ***	↑ ***
DC11c+ DCs	↑ ****	↑ ****	↑ ****

Arrows show the type of the changes (increase: ↑ decrease: ↓ no significant changes: n.s.), and asterisks indicate statistical significance (* *p* < 0.05, ** *p* < 0.01, *** *p* < 0.001, **** *p* < 0.0001).

**Table 3 cancers-11-01324-t003:** The investigated genes and the list of primer sequences used for the amplification.

Gene Name	Abbreviation of Gene Name	Primer Sequences
Hypoxanthine Phosphoribosyl-transferase 1	HPRT1	F: 5′ TCAGGCAGTATAATCCAAAGATGGT 3′
		R: 5′ AGTCTGGCTTATATCCAACACTTCG 3′
		P: 5′ CGCAAGCTTGCTGGTGAAAAGGACCC 3′
Damage Specific DNA Binding Protein 2	DDB2	F: 5′ GTCACTTCCAGCACCTCACA 3′
		R: 5′ ACGTCGATCGTCCTCAATTC 3′
		P: 5′ AGCCTGGCATCCTCGCTACAACC 3′
Growth Arrest and DNA Damage Inducible Alpha	GADD45	F: 5′ CTGCGAGAACGACATCAAC 3′
		R: 5′ AGCGTCGGTCTCCAAGAG 3′
		P: 5′ ATCCTGCGCGTCAGCAACCCG 3′
Sestrin 1	SESN1	F: 5′ GCTGTCTTGTGCATTACTTGTG 3′
		R: 5′ CTGCGCAGCAGTCTACAG 3′
		P: 5′ ACATGTCCCACAACTTTGGTGCTGG 3′
Ferredoxin reductase	FDXR	F: 5′ GTACAACGGGCTTCCTGAGA3′
		R: 5′ CTCAGGTGGGGTCAGTAGGA 3′
		P: 5′ CGGGCCACGTCCAGAGCCA 3′
Murine double minus-2 proto-oncogene	MDM2	F: 5′ CCATGATCTACAGGAACTTGGTAGTA 3′
		R: 5′ ACACCTGTTCTCACTCACAGATG 3′
		P: 5′ CAATCAGCAGGAATCATCGGACTCAG 3′

HPRT1 as housekeeping gene. Abbreviations: F—forward primer, R—reverse primer, P—probe.

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
