# Peer review of "Radiotherapy-Induced Changes in the Systemic Immune and Inflammation Parameters of Head and Neck Cancer Patients"

_cancers, 2019, doi:10.3390/cancers11091324_

Round 1

Reviewer 1 Report

  The authors described radiotherapy induced systemic immunosuppressive changes in the head and neck patients by analyzing peripheral blood samples. Several gene expression patterns were investigated concerning immune and inflammation-related changes.

Although I agreed their conclusion that radiotherapy made a systemic immune suppression in head and neck cancer patients, I think that this study had serious problems in the study design such as selection of the patients and so on.

  Firstly, sites of the tumors in their patients were not described and defined in their text and tables. As you know, head and neck cancer had various aspects in their own primary sites and the patients showed the different reactions such as mucositis or dermatitis induced by radiotherapy. For example patients with laryngeal cancer showed different patterns of mucositis and dermatitis from those with pharyngeal cancer. I think they must analyze and compare their results according to the classified sites, respectively. 

  Secondary, total doses of the radiation therapy were much various in their patients (51~68 Gy). They must select and arrange the patients who underwent radiotherapy with more narrow ranged total doses. If it was not possible, they must analyze the obtained data according to the doses of radiation therapy their patients received.

  Thirdly, size of the tumor (TNM classification) in their patients were not described and defined in their text and tables. As staging of the tumor influenced the fields and total doses of radiation therapy, it must be described accurately. I strongly recommend that this study must select and arrange the patients with head and neck cancer almost the same stage. For example, unifying the patients with T2N0 pharyngeal cancer. If it was not possible, they must analyze the obtained data according to the TNM classifications.

Reviewer 2 Report

In this study, the authors have investigated the changes in the immune system and inflammatory response in peripheral blood of 23 head and neck cancer (HNSCC) patients immediately in response to radiotherapy (RT), and one month post-treatment. This is important as changes in these parameters could potentially lead to the identification of predictive biomarkers of radiotherapy response. It was found that gene expression of fdxr, ddb2, mdm2 and gadd45 increased immediately post-RT, whereas sesn1 decreased. Levels of gadd45, mdm2 and sesn1 were maintained at these levels in blood samples 1 month post-irradiation. In relation to immune and inflammation-related plasma proteins, endoglin and ApoA1 were found to be significantly reduced in HNSCC patients which was exacerbated by RT, and the fractions of CTLA-4-expressing CD4 cells and PD-1 expressing Teff  cells were significantly higher in patients versus healthy controls. This suggests immune suppression in HNSCC patients, which is important given the increasing interest in immunotherapeutic agents in combination with RT for tumour control.

The manuscript is generally well written, the data are accurately described and will be of interest to the field of research. I have a few suggestions/comments.

Specific comments:

1.  Do the authors have the p53 status (wild-type or mutant) of the patient cohort to present, as well as HPV status (presumably most, if not all, are HPV-negative)?

2.  There is no correlation of the findings of gene expression and immune or inflammation proteins with patient response to treatment, in particular patients with no response or recurrence (e.g. 23, 31, 3). Whilst it is appreciated that these are low numbers, do the patients display a specific gene expression/protein profile? At least some comment in the Results/Discussion would be beneficial.

3.  The mRNA expression changes of selected genes in blood samples (Figure 1) is only from HNSCC patients pre- and post-irradiation. In contrast, the remaining figures have data from healthy controls, so the inclusion of this would help to understand if gene expression is generally different in the HNSCC patients.

4.  Data in Figure 2 should be expanded to include C5/C5a, CSCL5 and TFF3 protein concentration for complete presentation of the 8 proteins analysed, and to complement the summary of changes in Table 2.

5.  Increased CD4 proliferation was observed to be evident in only 40 % of the patients post-RT (Figure 4A). Do the authors have any additional details on the link of this with the clinical parameters of the HNSCC patients, or with CTLA4/PD-1 expression?

6.  Line 74, this sentence should be clear that radiotherapy, particularly low-LET photons, lead to isolated DNA damage and that the frequency of complex DNA lesions (defined as lesions generated in close proximity) is low.

7.  Improvements in grammar needed. For example, line 72 “…has been recognised for a long time”. Line 335, “These gene expression changes can manifest on directly irradiated cells…”.

Reviewer 3 Report

This manuscript by Katalin el al discusses the impact of chemoradiotherapy on inducing the immune and inflammatory response in head and neck cancer patients. Authors have discussed how such parameters can be used as a biomarker to classify patients based on the immune and inflammatory response. Authors have identified the differential expression of some proteins. The study design is very good and the authors have presented his work in a well-organized way. However, I have few suggestions.

1)      Head and neck tumors are the sixth most common …….absence of early symptoms [1].>> Give more reliable source for such information. References which include the cancer statistical information like American Cancer Society or similar international body is more reliable. Head and neck tumor is not the sixth most common cancer worldwide.

2)      Abstract: which will help in therapy individualisation and identification of patient subgroups who will benefit of certain therapeutic combinations> This part in the conclusion is too strong. Authors have data only from the 23 patients and they have not shown the real benefit of using identified proteins as biomarker so such conclusions should be presented as possibilities and not the real outcome. Authors should also possibly discuss the limitation of this study after conclusions or discussion where authors should point out the small sample size. The strong conclusions at the end of the manuscript should also be modified to make this study as a proposal for further investigation rather than declaring the discovery of biomarkers.

3)      In a pilot experiment> Avoid giving data from a pilot experiment in the manuscript. Only those data should be presented when finding is solid and there is no doubt. If authors believe data is right then don’t call it a pilot experiment.

4)      Around 80% of HNSCC cases present downregulation of 324 p53, either via TP53 mutation > in this sentence author have not talked about the role of HPV infection and E7 protein in regulating the TP53.

5)      Although authors have discussed the ethical clearance and patient's written content but nowadays a separate section in methods is preferred for this purpose.

 Minor

1)      detectable 1 month> one month

2)      will benefit of certain therapeutic combinations.>>check grammar

3)      1 month after the completion> one month. Similarly, changes need to be made at other places like 1 patient, 2 patient., 1 month, 5 genes,

4)      Although authors have talked about using proteins as biomarker found in plasma in the discussion, it may be of interest to readers if authors can discuss a little bit about plasma circulating DNA  as biomarkers (eg PMID: 28801876, 29763978, 31088830,).

5)      Authors have not mentioned within what time frame blood samples were processed.

6)      shipped to Hungary> for what purpose? Purpose should be mentioned wherever shipping is mentioned.

Reviewer 4 Report

Table 2 and 3.  Why is there an indication of a decrease or decrease in proteins or cells when there are no significant difference indicated in the data. Arrows should only be placed in Table 2 and 3 when there are significant differences. When there are no significant differences there should be a dash as indicated in the figure legend. More specifically, in Table 2 arrows show decrease in Adiponectin in patients before and directly after RT, but figure 2 does not show any significant difference (only an increase in patients 1 month after RT). The only proteins in the plasma of cancer patients that are shown (in Figure 2) to have a significant difference to healthy controls’ plasma are ApoA1 and Endoglin. Why are there arrows for the other proteins?

It is noted in the material and methods that blood was collected up to 6 hour after the last fraction but it would be useful for the reader to have “immediately after” or directly after” radiation treatment to be better defined in the text (line 173) or in the figure legend.

Round 2

Reviewer 1 Report

I think the authors have revised their manuscript according to my suggestions and requests thoroughly.  The readers would get certain information about immune and inflammatory circumstances in the patients with head and neck cancer who underwent radiotherapy.